# Demonstration of Hybrid Effect in Single Fiber Pull-Out Tests for Glass/Cellulose-Reinforced Polypropylene with Different Fiber–Matrix Adhesions

**DOI:** 10.3390/polym14132517

**Published:** 2022-06-21

**Authors:** Christian Kahl, Julius Bagnucki, Jan-Christoph Zarges

**Affiliations:** Institute of Material Engineering, Polymer Engineering, University of Kassel, 34125 Kassel, Germany; julius.bagnucki@gmail.com (J.B.); zarges@uni-kassel.de (J.-C.Z.)

**Keywords:** hybrid reinforcement, fiber pull-out test, fiber–matrix adhesion, UV-light treatment

## Abstract

In hybrid fiber reinforcement, the combination of glass and regenerated cellulose fibers is a promising combination because the different properties of the fibers can be combined. The properties of the regenerated cellulose fiber in combination with the absorption of energy by fiber pull-outs can thus significantly increase the toughness of the composite in the event of failure, while the glass fiber significantly increases the stiffness and strength due to its properties. In this study, the interaction of the two fiber types in a composite is demonstrated by fiber pull-outs. For this purpose, the fibers are embedded in a PP matrix and simultaneously pulled out. Different bondings of the fiber by, e.g., coupling agent and/or a pretreatment of the regenerated cellulose fiber, were also investigated. The results show that each type of fiber has a characteristic force–deformation curve, and the hybrid reinforcement is a combination of both curves. The use of a coupling agent leads to an increase in the interfacial shear stress from 4.5 to 7.5 MPa. A treatment of the regenerated cellulose fiber by UV light further increases the interfacial shear stress to 11 MPa.

## 1. Introduction

With regard to hybrid fiber reinforcement in combination with thermoplastics, the combination of glass fibers (GF) and regenerated cellulose fibers (RCF) is a popular combination. The interaction of these two fiber types has already been illustrated in short fiber-reinforced polypropylene (PP) [1,2,3,4]. On the one hand, the GF contributes to high strength and high resistance to deformation; on the other hand, a tough RCF counteracts the brittle behavior of glass fiber-reinforced thermoplastics [5,6]. In addition to the mechanical properties, a substitution of GF with a sustainable RCF is another reason for this fiber-type combination [7]. RCF is produced in the viscose process and is based on natural resourced cellulose made from soft wood. Compared to other natural fibers, RCF have higher thermal resistance and higher ductility [8]. In addition to the properties of the two fiber types, the appearance is also different. The GF appears in a cylindrical form and is hydrophobic. Even the process temperatures do not affect the shape of the GF. RCF, on the other hand, has folds along the fiber and is hygroscopic [9].

Nguyen et al. have studied pull-outs of hemp fibers from PP and have reported that both fiber diameter and embedded fiber length have an effect on interfacial shear strength (IFSS). Furthermore, the process temperatures during fiber embedding were also investigated, and a temperature of 183 °C was found to be optimal. Temperatures below and above this resulted in a decrease in IFSS [10].

The interface between GF as well as RCF and PP has also already been investigated in single fiber pull-out tests [11]. The IFSS during the pull-out of a single fiber depends on the interface between the fiber and the matrix and can be influenced by different treatment methods or coupling agents [10,11,12]. For RCF, it has been shown that the polar part of the free surface energy can be increased by UV light treatment [13,14]. In combination with a polar matrix such as PBT, the IFSS after fiber treatment is so high that there is no fiber pull-out from the matrix but fiber breakage occurs. In combination with PP, UV light treatment is only effective if a coupling agent is mixed into the matrix to which the polar groups of the UV-treated fiber can bind [10,15,16].

The use of a coupling agent was investigated by Sergi et al. for a hybrid fiber reinforced PP with cellulose and basalt fibers. The use of an anhydride-grafted PP led to an increases in tensile, flexural and impact properties. At the weight percentages of 15% basalt and 15% of RCF, the Young’s modulus was increased by 284% and the flexural modulus was increased by 263% compared to neat PP [17]. 

The resistance to deformation can be seen in a force–deformation curve for single fiber pull-outs and will be investigated with the combination of GF and RCF in this publication. Zarges et al. also carried out single fiber pull-outs for GF or RCF in PP. With the IFSS, critical fiber lengths were calculated and compared with the fiber length distribution in the injection-molded composites. The results showed that in a GF-reinforced compound, the fibers are shortened to such an extent that almost no fibers can be stressed to breakage by the load transfer of the matrix. In RCF, it is just 4% of the fibers that theoretically break when the composite fails [9]. For this reason, this study demonstrates the interaction of GF and RCF through fiber pull-outs.

The interaction of the two fiber types GF and RCF could already be investigated in hybrid fiber-reinforced composites [3], in which the fiber pull-outs were carried out with only one GF or RCF. In this publication, the interaction of the two fiber types in the fiber pull-out is to be demonstrated by embedding two fibers next to each other in a polypropylene matrix and pulling them out simultaneously. Since the shell surface of the GF and RCF can be considered the same, a direct comparison can be made [9]. The reference samples were made with two fibers of one type for comparison. A coupling agent was added to the matrix to bond both the RCF and GF more strongly to the matrix. The UV-light treatment of the RCF will also be investigated to show the possibility of bonding only one fiber more strongly in a hybrid reinforced system. The treatment of a natural fiber can lead to a different bonding between fiber and matrix by creating free radicals on the surface of the fiber [18].

## 2. Materials and Methods

### 2.1. Materials

A polypropylene Type PP 520P provided by Sabic (Duesseldorf, Germany) was used for the investigations. The density of this polymer is 0.905°g/cm^3^, and the melt volume rate is 10.5 g/min at 230 °C/2.16 kg. Properties of the matrix can be found in Table 1. The improvement of the fiber-matrix interface was carried out with a coupling agent MAPP 6452 in this study, which is a maleic anhydride grafted PP wax provided by the company Clariant AG (Muttenz, Switzerland). The coupling agent was mixed to the PP granulate before film extrusion with a content of 2 wt%.

Individual fibers extracted from a roving made of GF and RCF were used for the fiber pull-out tests. Data on the density and mechanical properties of the rovings can be found in Table 1. The E-glass roving type 490 is manufactured by Johns Manville and was purchased from Mühlmeier GmbH&Co.KG in Bärnau, Germany. The roving made of E-Glass has a soft silane coating on the surface, which promises good bonding to polypropylene.

The regenerated cellulose fiber type CR250 is manufactured by Cordenka GmbH in Obernburg, Germany using the viscose process. It promises a consistent chemical structure and reproducible properties compared to natural fibers. Since the regenerated cellulose fiber is hydrophilic, the fibers were heat treated for 10 min at 210 °C before they were embedded in the matrix to ensure a low moisture content. The temperature was chosen because similar melting temperatures prevail in PP processing [19], and thermal degradation of the cellulose structure can be largely excluded at these temperatures, as shown in several publications [20,21].

**Table 1 polymers-14-02517-t001:** Properties and parameters of used material [10,22].

Roving Type	Fiber Diameter (µm)	Number of Fibers	Density of Material (g/cm^3^)	Titer (g/km)	Youngs Modulus (MPa)	Tensile Strength (MPa)	Elongation at Break (%)
E-glass fiber	15.5	1250	2.54	600	73,000	3400	3.5
Cordenka Cellulose CR-Type	12	1350	1.5	244	2200	825	13
Sabic PP 520P	-	-	0.905	-	1700	36	-

### 2.2. Film Extrusion

For the fiber pull-outs, films were produced that were needed to embed the fibers in the matrix. For this purpose, films made of PP and PP/2%MaPP were produced on a calender (Chill Roll CR 136/350), provided by the company Dr. Collin (Ebersberg, Germany). A single screw extruder type TR 14/24 GM, provided by the company Gimac (Italy), placed in front of the calender plasticizes the granulate and was set to a temperature of 200 °C. The screw speed was set to 100 rpm. A slit die with a width of 30 cm was connected to the extruder. The thickness of the extruded material can be adjusted via flex lips. The temperature of the slot die tool was set to 200 °C. On the calender, the melt was cooled and stretched over rolls until a desired thickness of approx. 100 µm was reached. The production of the films was manufactured according the publication of Zarges et al. [9].

### 2.3. UV-Light Treatment

Single continuous fibers were extracted out of the rovings to be treated with UV-light. Due to the chemical structure, the treatment of RCF with UV light is promising with regard to the formation of free radicals. An effect of UV light on GF could not be demonstrated so far. It has already been shown that treatment with high-energy light on RCF leads to degradation of the fiber, while the degradation increases with increasing treatment time. The light intensity decreases with raising distance from the light source [10]. For this reason, a short treatment time of the RCF of 10 s at a distance of 2 cm to the light source was chosen. Activation of the surface should demonstrate a strong surface of the RCF to a PP with coupling agent [23].

Four lamps with emitting UV-C light are used as the light source (Figure 1). The lamps have a length of 30 cm each and a power of 24 W. They emit wavelengths in the UV-C range with a high concentration of wavelengths at 185 nm and 254 nm. The ozone formed at 185 nm can react with the free radicals of the activated surface and increases the concentration of oxygen on the surface of the fiber. The higher concentration of oxygen leads to stronger bonding of the fiber to the matrix [10,24].

### 2.4. Embedding Fibers

The fibers were embedded in the matrix using a mold (Figure 2) and a hot press 300 K-70-L, provided by Paul Ott (Lambach, Austria) [9]. Therefore, fibers were extracted from the roving, and their length was shortened so that they stuck out of the mold on both sides. The temperature in the press was set to 190°C and the pressure was set to 2 N/mm^2^. After reaching the temperature of 190 °C, the press was opened after 30 s to remove the sample. The temperature of 190 °C has no significant effect on the RCF in terms of degradation, which has already been shown by Feldmann et al. [20]. The samples were then taken at the point where the fiber stuck out of the matrix. The fibers were positioned close together so that both fibers could be pulled out simultaneously during the fiber pull-out test.

### 2.5. Characterization

The hybrid effect was demonstrated on fiber pull-outs by comparing the maximum IFSS of the reference samples and the hybrid-reinforced samples (Figure 3). In this case, the IFSS is divided by the surface area of both fibers.

### 2.6. Fiber Pull-Out Test (FPT)

After the fiber was embedded in the matrix, the fiber was pulled out with the single fiber tensile testing system Favimat+, provided by the company Textechno (Moenchengladbach, Germany). The test speed when pulling out the fiber was reduced to 0.5 mm/min, since the RCF breaks at high test speeds and is not pulled out. A higher test speed resulted in many fiber breaks, which cannot be used for an evaluation. The FPT were carried out using at least 7 samples for each combination. The references were made with two fibers of the same type so that the fiber sheath area was as similar as possible for all samples. Zarges et al. showed that the cross-section of the RCF is decreased after a thermal treatment by only about 1% [9]. In addition, the embedding length of the fibers was kept as constant as possible.

The maximum force (*F_max_*) during fiber pull-out was used to calculate the IFSS. It must be divided by the diameter of the fiber (*d_f_*) and the length of the fiber embedded in the matrix (*l_ef_*) (see Equation (1)). The IFSS is calculated using the equation according to Kelly and Tyson [25]:(1)$$\tau =\frac{F_{max}}{d_{f}*\pi *l_{ef}}$$

The maximum force is higher for a pull-out of two embedded fibers than for a single fiber, because the surface area of two fibers is approximately twice as high. In Formula (1), the force must then be divided by the surface area of both fibers:(2)$$\tau_{hyb}=F_{max}\left(d_{f(1)}*\pi *l_{ef(1)}\right)+*\pi *l_{ef\left(2\right)})$$

The pretension of the fibers clamped in the Favimat+ must be as equal as possible before closing the clamp. For this reason, the film is held in the upper clamp and the fibers hang down with a clamp on each fiber that initiates an almost equal preload due to its weight (see Figure 4).

After the fibers were pulled out, the film was examined under a microscope to determine the embedded fiber length *l_ef_*. A digital microscope VH-Z 100 provided by Keyence GmbH (Neu-Isenburg, Germany) was used for this purpose. Images were made with a magnification of 1000×. The films were examined with incident light. In the microscope’s software, the distance between two points can be measured with a distance tool, which was useful for measuring *l_ef_*. The transparency of the film made it easy to see and measure the empty channels of the pulled-out fibers, as can be seen in Figure 5. Due to the similar perimeter of the fiber types, the surface area of the embedded fiber can be compared well.

## 3. Results and Discussion

This chapter shows the results of the research performed in this study. The results are discussed based on the theses from the introduction and related publications.

### 3.1. Fiber Pull-Out Test (FPT)

Fiber pull-outs were performed on reference samples of two fibers of the same type and on hybrid reinforced samples with one GF and RCF each. Different fiber bonds were also considered, which lead to a change in the IFSS. For the tests, at least seven samples per fiber combination were tested. Figure 6 shows the results in force–deformation curves on fibers with no UV-light treatment or coupling agent. As expected, the reference specimen with two GF shows the highest maximum force because of a silane based surface that promises a good adhesion to PP. The maximum force already occurs at a low deformation of 5.6%. At just over 20 N, the maximum is reached, and the fibers begin to be pulled out of the matrix. In the reference sample with two RCF, the rise of the curve is less steep than in the GF reference sample and reaches its maximum at just under 15 N and a deformation of 30%. For both the RCF-RCF and the GF-RCF specimens, a sudden flattening of the curve up to the maximum force can be seen at 80% of the maximum force. This behavior cannot be seen with the GF-GF specimen. The progress of the fiber pull-out after the maximum force is essentially dependent on the effective embedded length of the fiber (*l_ef_*).

Zarges et al. have shown the force–elongation curve of individually embedded fibers in PP. The curves also show that GF in combination with PP has a high maximum force with low deformation compared to RCF, while RCF in combination with PP has a comparatively low maximum force with significantly higher deformation [9]. Figure 6 thus shows the interaction of the two fiber types, since the course of the GF-RCF sample up to the force maximum reflects a combination of the GF-GF and RCF-RCF reference samples.

The interaction of the fibers was also investigated with different bonding of the fibers to the matrix. The difference between the bonds is that in the first step, the coupling agent is mixed into the matrix material which influences the bonding of both fiber types. In a second step, only the RCF was treated with UV light to reach a further improvement of the bonding between RCF and matrix. In this sample, the GF has a better bonding only because of the coupling agent and was not treated with the UV-light. The results of the different fiber bonds show a change in the force–deformation curve (Figure 7).

The black curve in Figure 7 shows the neutrally connected fibers to the PP matrix with the course already described. A stronger bonding of the fibers by the coupling agent (red and blue curve) leads to the fact that the resistance of the specimen against a fiber pull-out is just as steep, but the maximum force to initiate the fiber pull-out is higher, and a leveling of the curve due to the tough RCF is no longer visible. The coupling agent increases the maximum force from 15 to 17.5 cN. However, the maximum of the curve is also reached at a lower deformation of only 10%.

The effect of a coupling agent has already been demonstrated in previous publications. The stronger adhesion of the fiber to the matrix also increases the maximum force. The silane-based sizing on the surface of the GF interacts with the functional groups of the coupling agent and leads to a stronger bonding that results in higher IFSS. The functional groups of the coupling agent also interact with the surface and functional groups of the RCF. Due to the effect of the coupling agent on both fibers of the GF-RCF sample, the maximum force increases, but the interaction of the two types of fibers by a high maximum force at higher deformation is no longer visible. The stronger bond of the RCF results in the toughness of the fiber playing no further role in the force/deformation curve, and the RCF only contributes to a high maximum force. This effect is enhanced by an additional UV light treatment of the RCF. The UV treatment influences the maximum force during pull-out and increases it from 17.5 to 24 cN. The maximum force occurs at the same deformation of 10%.

### 3.2. Interfacial Shear Strength (IFSS)

The results of the reinforced polypropylene films with two fibers are shown in Figure 8. For the two reference samples, the IFSS of the GF sample is the highest, as expected. The IFSS of the RCF sample is 4.5 MPa and can be increased by 25% (5.7 MPa) by the interaction of the GF and RCF. The high stiffness and strength of the GF leads to higher maximum force values compared to the reference with two RCFs, resulting in higher shear stresses according to Formula (2). The high force transmission of the GF compared with the RCF is on the one hand based to the silane-based sizing on the outer surface of the fiber and on the other based on the properties of the fiber. The sizing of the GF promises a strong bonding of the fiber to the PP.

The use of a coupling agent has been demonstrated several times in the literature. With the interaction of the fibers, the IFSS can be increased by 31% compared to the sample without a coupling agent. In this case, the coupling agent has an effect on both fiber types and leads to a stronger bonding of the GF and RCF to the PP. The result is a higher maximum force leading to an increase in IFSS compared to the PP+RCF-GF sample. The effect of UV light on RCF in combination with a coupling agent has already been demonstrated in a previous publication [10]. In these investigations, the IFSS can be further increased by the UV treatment of the RCF. This effect can also be observed in this publication. UV treatment of the RCF leads to the formation of free radicals, which can react with the coupling agent and result in stronger bonding of the RCF to the PP+MAPP. Due to this effect, the IFSS can be increased from 7.5 to 11 MPa.

## 4. Conclusions

The experiments for this publication were carried out to demonstrate the interaction of the two fiber types in a PP matrix by fiber pull-out test. Different fiber bondings to the matrix through a coupling agent and UV-light treatment methods of RCF were also investigated. The results of the fiber pull-outs lead to the following conclusions.

The interaction of the two fiber types, GF and RCF, embedded side by side in a PP matrix can be demonstrated by fiber pull-outs. Due to the properties of GF in a hybrid reinforced PP+RCF-GF sample, the force/deformation curve increases steeply and reaches a higher force maximum compared to the RCF reference sample. The toughness of the RCF in the hybrid reinforced specimen results in the force maximum being reached at a high deformation, thus combining the properties of both fiber types.The use of a coupling agent affects both the bonding of the GF and the RCF. The interaction of the functional groups of the coupling agent with the surface of the fibers and the matrix leads to a stronger bonding of the fiber to the matrix compared to a pure PP without coupling agent. The result is an increase in the maximum force in the force-deformation curves, which also leads to higher IFSS. However, due to the changed bonding of the RCF, the force/deformation curve also loses the toughness of the RCF. The maximum force is already reached at low deformations in the hybrid reinforced specimens with coupling agent, as can also be seen in the GF reference specimens.As already shown in the single fiber pull-outs, a treatment of the RCF with UV light leads to stronger bonding of the fiber in combination with a coupling agent. The UV treatment of the RCF results in the formation of free radicals on the surface of the fiber, which bond with the coupling agent. This results in a stronger bond of the RCF to the PP compared to the PP and the adhesion promoter. The particularly strong bonding of the RCF leads to an increase in the maximum force in the force/deformation curves and, respectively, to a higher IFSS. Due to the strong bonding, the toughness of the RCF in the force–deformation curves is also lost after UV treatment, and the maximum force is already reached after small deformations.The UV treatment can influence the bonding of the RCF to the matrix. In combination with a coupling agent, the fiber can be bonded even more strongly, and thus, the IFSS can be increased. With a coupling agent, both fiber types are more strongly bonded. A selective interface in a hybrid fiber reinforced composite on only one fiber type is therefore possible with UV light.

## Figures and Tables

**Figure 1 polymers-14-02517-f001:**
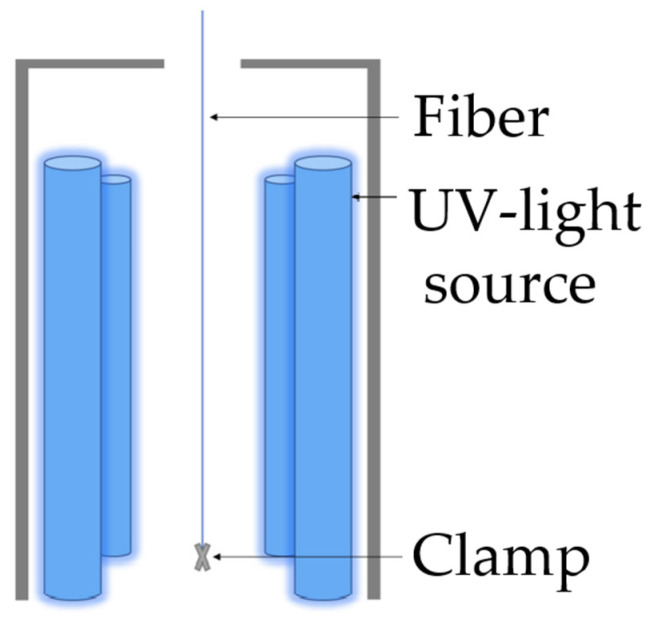
Schematic treatment of RCF with UV light.

**Figure 2 polymers-14-02517-f002:**
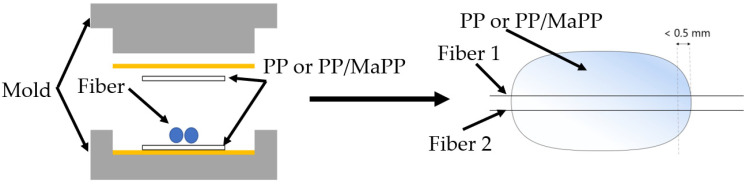
Tool for embedding the fibers between two matrix films (**left**) and a sample with embedded fibers (**right**).

**Figure 3 polymers-14-02517-f003:**
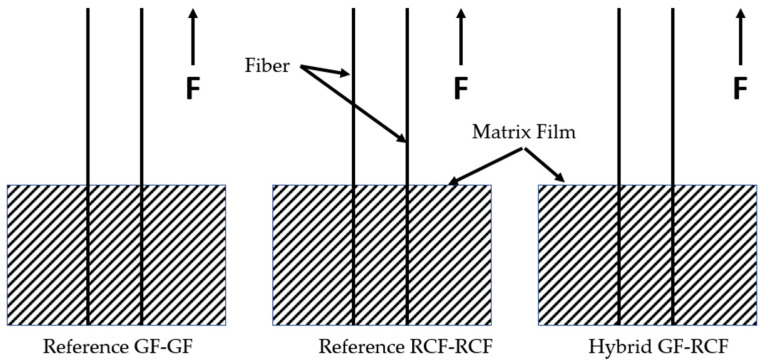
Schematic representation of the reference specimens and the hybrid fiber-reinforced specimen for fiber pull-out.

**Figure 4 polymers-14-02517-f004:**
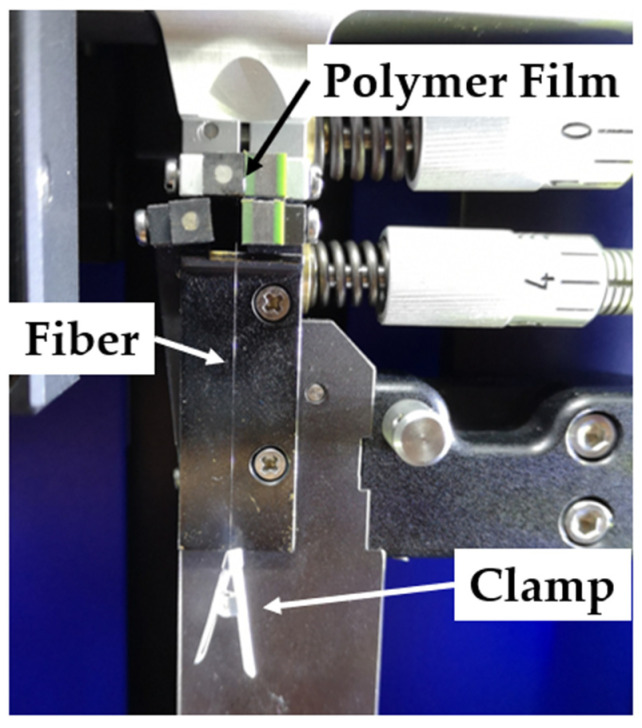
Setting up a fiber extract sample in the Favimat+.

**Figure 5 polymers-14-02517-f005:**
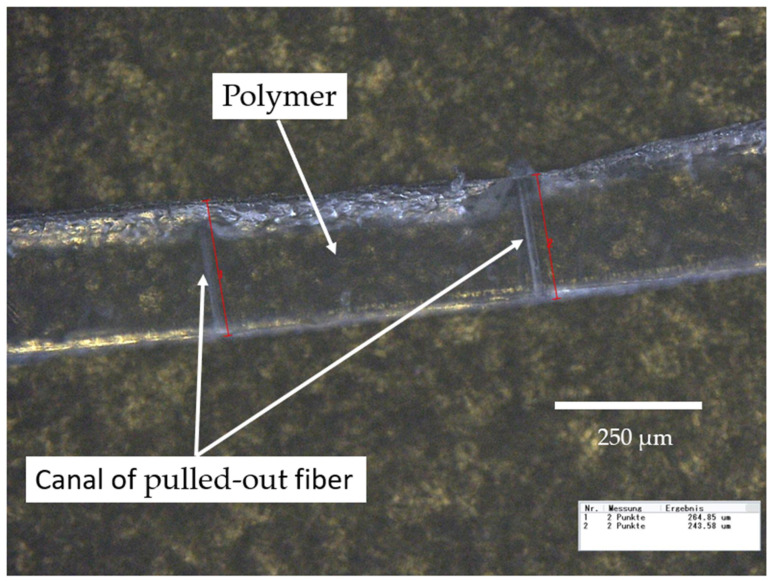
Image of polymer film with canals of pulled-out fiber.

**Figure 6 polymers-14-02517-f006:**
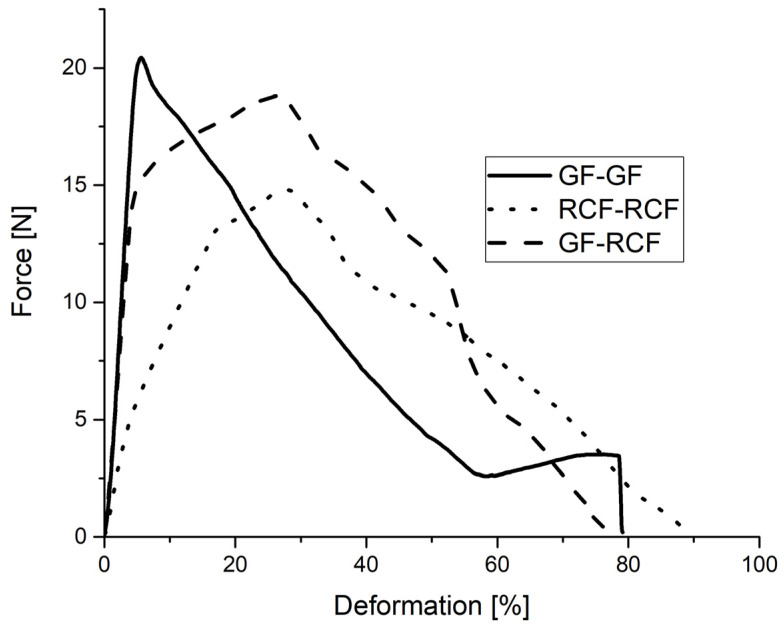
Deformation/Force course of hybrid and reference samples in fiber pull-out tests.

**Figure 7 polymers-14-02517-f007:**
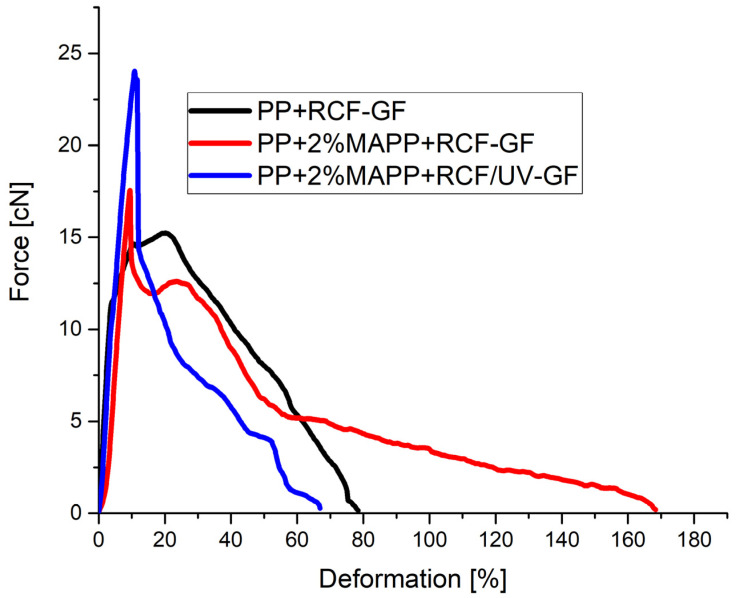
Force–Deformation of hybrid reinforced PP with different fiber bonding.

**Figure 8 polymers-14-02517-f008:**
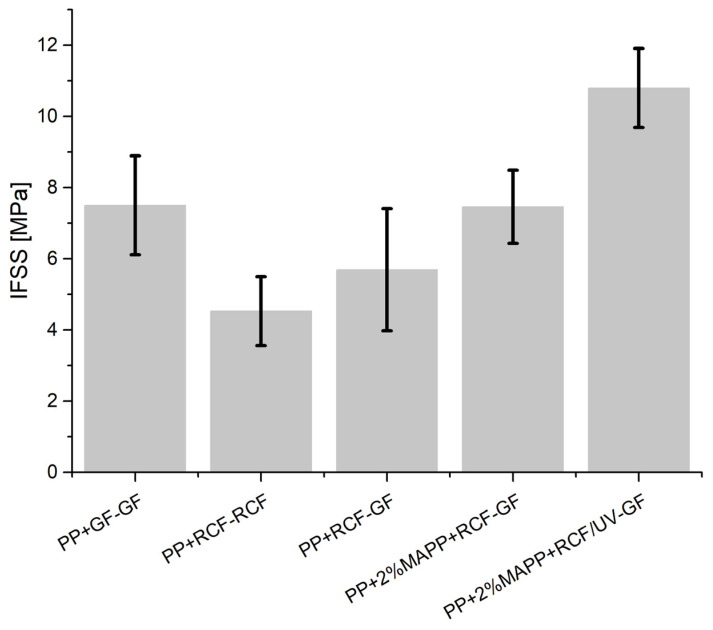
IFSS of hybrid amplified samples with reference values and different bonding.

## Data Availability

Not applicable.

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
