# Peer review of "Demonstration of Hybrid Effect in Single Fiber Pull-Out Tests for Glass/Cellulose-Reinforced Polypropylene with Different Fiber–Matrix Adhesions"

_polymers, 2022, doi:10.3390/polym14132517_

Round 1

Reviewer 1 Report

Demonstration of hybrid effect in single fiber pull-out tests for glass/cellulose reinforced polypropylene with different fiber-matrix interfaces 

Christian Kahl, Julius Bagnucki, Jan-Christoph Zarges

This work is too technologically oriented and would benefit from a more theoretical approach.

However, the measurement results are precisely proceeded and the discussion of the results is just as good. I propose to accept this article without changes.

Author Response

Thank you for your review. 

Reviewer 2 Report

Thank you for submitting your paper. The work done here draws attention to a significant subject performance of glass/cellulose reinforced polypropylene and hybrid fibre reinforcement. I have found the paper to be interesting. However, several issues need to be addressed properly before the paper is being considered for publication. My comments including major and minor concerns are given below:

Please consider reviewing the abstract and highlight the novelty, major findings, and conclusions. I suggest reorganizing the abstract, highlighting the novelties introduced. The abstract should contain answers to the following questions:

What problem was studied and why is it important?

What methods were used?

What conclusions can be drawn from the results? (Please provide specific results and not generic ones).

The abstract must be improved. It should be expanded. Please use numbers or % terms to clearly shows us the results in your experimental work.

Please consider reporting on studies related to your work from mdpi journals.

The introduction must be improved, it is too short, provide more in-depth critical review about past studies similar to your work, mention what they did and what were their main findings then highlight how does your current study brings new difference to the field.

The authors should add a list of nomenclature for all the Greek letters and symbols used in the study.

Remove the numbering from the keywords, it is unnecessary.

Line 31 and any other places: remove bulk citations unless each citation is given full credit.

Combine all small paragraphs which are less than 5 lines together in order to create larger paragraphs and improve the readability and flow of the manuscript.

Reduce font size in Table 1

Avoid using titles in subsections such as 2.2 Experimental. The authors need to include meaningful captions

Combine figures 1 and 2 into one figure and use (a) and (b) instead.

Combine figures 3 and 4 into one figure and use (a) and (b) instead.

Figure 4 add scale bar.

Mention all test standards used in the current study in section 2.

The size of the study is small, how can the authors justify that?

Line 210, it is not clear why the authors add a reference at the end of this line.

The data in figure 5: how many samples each were used to generate this graph? If 1-2 samples each, then this is not enough, and additional tests should be carried out to support the results.

 Images of fabricated samples, equipment used for testing should be included in the materials and methods section.

Line 243 does not read well, please consider rephrasing it.

Line 242, in that case what is the recommended UV treatment after which it does not contribute to improve performance?

Overall, simple study and small size, could be expanded by analysing other factors/mechanical responses in order to obtain a wider picture of the effect of fibres interaction and fibre bonding.

The results are merely described and is limited to comparing the experimental observation and describing results. The authors are encouraged to include a more detailed results and discussion section and critically discuss the observations from this investigation with existing literature.

Author Response

Thank you for your review. I changed the publication due to your comments. Please see the changes marked in your comments attached and the changed manuscript. 

Reviewer 3 Report

Interesting and well-prepared article "Demonstration of the hybrid effect in monofilament drawing tests for glass fiber reinforced polypropylene / cellulose with different fiber-matrix interfaces". The authors carefully collected the information and interpreted the results of research into the properties of fiber-reinforced polypropylene composites. 

I recommend a minor correction before publication:

- Please complete the abstract with specific numerical values of the examined properties, e.g. maximum force F or IFSS,  

- Please complete the introduction with a few items from the last years 2020-2022, e.g.

* Hybrid Cellulose – Basalt Polypropylene Composites with Enhanced Compatibility: The Role of Coupling Agent, https://doi.org/10.3390/molecules25194384

* FTIR-ATR spectroscopic, thermal and microstructural studies on polypropylene-glass fiber composites, https://doi.org/10.1016/j.molstruc.2022.133181

* Swelling-based preparation of polypropylene nanocomposite with non-functionalized cellulose nanofibrils, https://doi.org/10.1016/j.carbpol.2021.118847

-Please specify the thermal stability of the regenerated cellulose fibers (RCF) and whether it does not interfere with the composite preparation process.

-Figure 4: Please add a clear scale to the photo

Author Response

(The authors gave the same response as above.)

Round 2

Reviewer 2 Report

The authors need to expand the introduction as indicated by previous review, adding 5 extra lines in not enough, also line 29 remove bulk citations.

What is the percentage of self citation in this work?
